# Strategic coastal dike shape for enhanced tsunami overflow reduction

**Naoki Takegawa** [1]*, **Yutaka Sawada**[2], **Noriyuki Furuichi**[1]

**1** National Metrology Institute of Japan (NMIJ), National Institute of Advanced Industrial Science and Technology (AIST), Tsukuba, Ibaraki, Japan, **2** Graduate School of Agricultural Science, Kobe University, Nada-ku, Kobe-shi, Hyogo, Japan

* takegawa-naoki@aist.go.jp

## Abstract

Coastal dikes are an essential social infrastructure to mitigate tsunami damage. However, there are no clear guidelines on effective dike shapes for reducing tsunami overflow. To examine effective dike shapes, numerical simulations of the amount of tsunami overflow at coastal dikes are conducted with reference to tsunami waveforms caused by the Great East Japan Earthquake. Results reveal the relationship between the dike shape and the amount of the overflow; the mechanism of overflow reduction based on the velocity and water level distribution is also verified. The comparison of the seaward and landward slopes of coastal dikes reveals that the seaward slope has a greater impact on the overflow, and the seaward slope with a vertical wall or a wave return structure reduces the overflow by 5%–30% compared to the 1:2 (26.6˚) seaward slope. The landward slope should be determined based on the tsunami scale and the scour related to the dike stability. Since tsunami inflow damages human life and social infrastructure, achieving the overflow reduction without increasing dike height is vital. Our work contributes to rational design guidelines for coastal dikes.

**Data Availability Statement:** The datasets are within the paper and Supporting Information.

**Funding:** This work was supported by JSPS KAKENHI Grant Number 21H02306 (S.Y. and T. N.). https://www.jsps.go.jp/english/e-grants/ The funder had no role in study design, data collection

## Introduction

A tsunami caused by an earthquake severely harms human life and social infrastructure. The 2004 Sumatra earthquake and the resulting tsunami caused the highest number of casualties since 2000 [1]. The 2011 Great East Japan Earthquake with magnitude 9.0 was the fourth largest in world history [2], and many coastal protection facilities such as seawalls and coastal dikes were destroyed by the resulting tsunami. Based on the Great East Japan Earthquake, the Cabinet Office mentions the following as a maximum class tsunami countermeasure [3]. "In preparing for largest-possible tsunami, raising considerably the designed tsunami height for the coastal protection facilities is not realistic from the standpoint of the financial requirements for construction of such facilities, and the potential impact on the coastal environment and its use." This means that, although the reduction in the amount of tsunami inflow to the landward side is extremely important for tsunami damage control, the overflow must be allowed for the largest class of tsunami. Mikami et al. [4], Bricker et al. [5], Tanaka et al. [6], Tokida and Tanimoto [7], and Tanaka and Sato [8] conducted a field survey on the Great East Japan

and analysis, decision to publish, or preparation of the manuscript.

**Competing interests:** The authors have declared that no competing interests exist.

Earthquake and pointed out the detachment of the covering blocks of coastal dikes and the scour on the landward ground of coastal dikes due to tsunami overflow as the cause of dike failures. The amount of tsunami inflow to the landward side increases dramatically due to the dike failure. Therefore, since the Great East Japan Earthquake, many studies have been conducted to prevent coastal dike failures by tsunami overflow. Important studies other than dike failure prevention include the reduction in the flow velocity, water depth and water energy in the landward area behind coastal dikes. Rahman et al. [9,10] proposed a method to reduce the velocity and energy of overflow water in the landward area by using the vegetation on a mound. Tanimoto et al. [11] pointed out that the presence of a landward channel behind a coastal dike reduces the velocity and inundation depth of overflow water. Clearly, the reduction in flow velocity and water depth in the landward area is important. In addition, since velocity × depth means the flow rate of overflow water, reducing the overflow itself is also an extremely important aspect in tsunami mitigation.

The studies described above provided insights into dike failure prevention, paving the way for the study of an effective dike shape to reduce tsunami overflow (**Fig 1**). As long as the overflow is allowed, the challenge for hydrodynamics in constructing coastal dikes is how to reduce the amount of the overflow. However, as tsunami has wavelengths that range from several kilometers to several tens of kilometers, performing an experimental verification is difficult. Estebsn et al. [12,13], Xu et al. [14] and Harish et al. [15] reported useful findings on wave forces and inundation heights by subjecting coastal structures of various shapes to bores in the experiments and numerical simulations. However, these studies did not focus on the amount of tsunami overflow. In addition, since the coastal structures studied remained at the laboratory scale and the waves used were bores, it would be desirable to conduct more detailed studies on the applicability at larger scales for long-period tsunami.

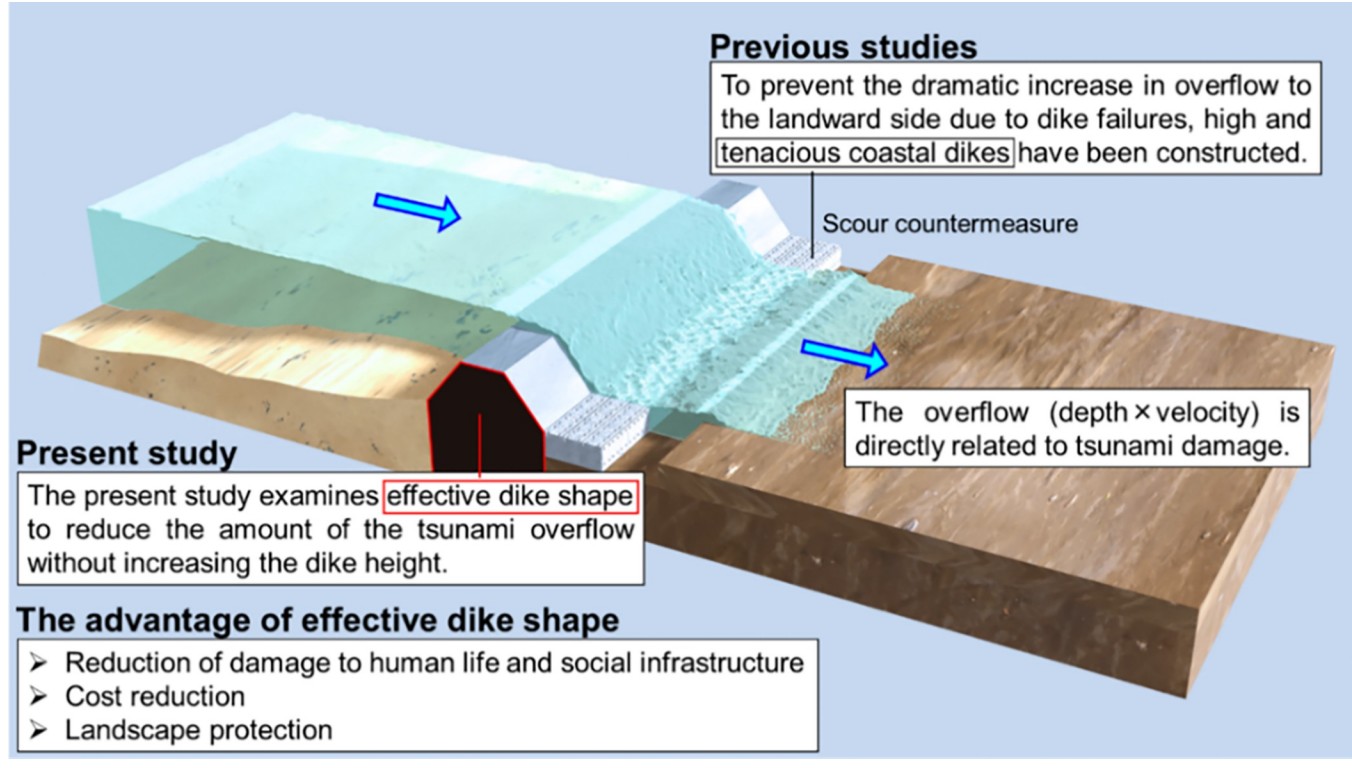

**Fig 1. Study of an effective dike shape to reduce tsunami overflow.**

To reduce tsunami inflow, this study uses numerical simulations to investigate the effective dike shape for overflow reduction. The unique feature of this study is that it focuses on overflow phenomena on a real scale (several kilometers) and examines various dike shapes based on the wave height data of the Great East Japan Earthquake and bathymetry data. This paper correlates the amount of the overflow with the dike shape using an innovative hydrodynamic approach, while current methods rely on increasing the dike height to reduce the amount of the overflow. In addition, the numerical simulation is compared with the experiments of Mitui et al. [16] to ensure the validity of the simulation and confirm the reproduction of the overflow phenomena.

## Materials and methods

### Numerical simulation conditions

In this study, numerical simulations of overflow phenomena on a real scale, which is difficult to reproduce experimentally, were conducted to investigate the effective dike shape for overflow reduction. The simulations were performed based on the information shown in **Fig 2**. To reduce the computational load, we used the observed data from the coastal wave meter (**Fig 2 (A)**) as the tsunami waveform because the distance between the meter and the shoreline was close and the period of the water level change was short. As shown in **Fig 2(B)**, the distance between the coastal wave meter and the shoreline is approximately 2300 m. This 2D plane section was used as the simulation domain. The tsunami waveform at the coastal wave meter, shown by the solid blue line in **Fig 2(C)**, was reproduced as linear waves (small-amplitude waves and Airy waves).

The simulation conditions are shown in **Fig 3**. Based on the information in **Fig 2(B)**, the water depth at the wave-generation boundary (**Fig 3(A)**) is set to 40 m, and the distance from the wave-generation boundary to the seaward slope shoulder of the coastal dike is fixed at 2310 m in all cases. The dike shapes considered in this study involve four seaward slopes (1:2 (26.6˚), 1:1 (45˚), Vertical and Circle), two landward slopes (1:2 and 1:1), and three dike heights (3, 5 and 10 m) for a total of 24 cases (**Fig 3(B)**). The seaward slope of Circle is assumed to be the wave-return structure.

Vertical landward slopes were excluded from consideration because of the possibility that the turbulence model used in this study (RANS $k$–$\varepsilon$ model) may not accurately reproduce the flow separation. Linear waves for half waveforms were given as input conditions. **Fig 3(C)** shows the water level change at $x = 5$ m near the wave-generation boundary. After the simulated half-period tsunami was generated (after 120 s), the flow velocity at the boundary is set to zero. Although the linear waves in the simulation do not perfectly reproduce the water level change of the coastal wave meter shown by the solid blue line in, they show approximately the same wave height and period. Note that it is not necessary to perfectly reproduce the actual tsunami waveform because the purpose of this study is to clarify the effective coastal dike shape for overflow reduction against a real-scale tsunami. For simplicity, the seafloor topography from the wave-generation boundary at a depth of 40 m to the dike location is represented as a straight line. On the right side of the simulation domain, the coastal dike and outlet boundary exist, and the total amount of the overflow was measured by integrating the instantaneous flow rate observed at the outlet boundary. The simulation time for each case was set to 300 s when the flow rate from the outlet boundary was sufficiently small. From the simulation, we obtained measurements such as water levels, flow velocities, and pressures at various times within the simulation domain. These measurements were then directly used to depict the flow fields around the coastal dike and the amount of the overflow. In deriving these results and verifying the dike shapes against the assumed conditions, we did not employ complex statistical methods. However, for future developments, especially when constructing design

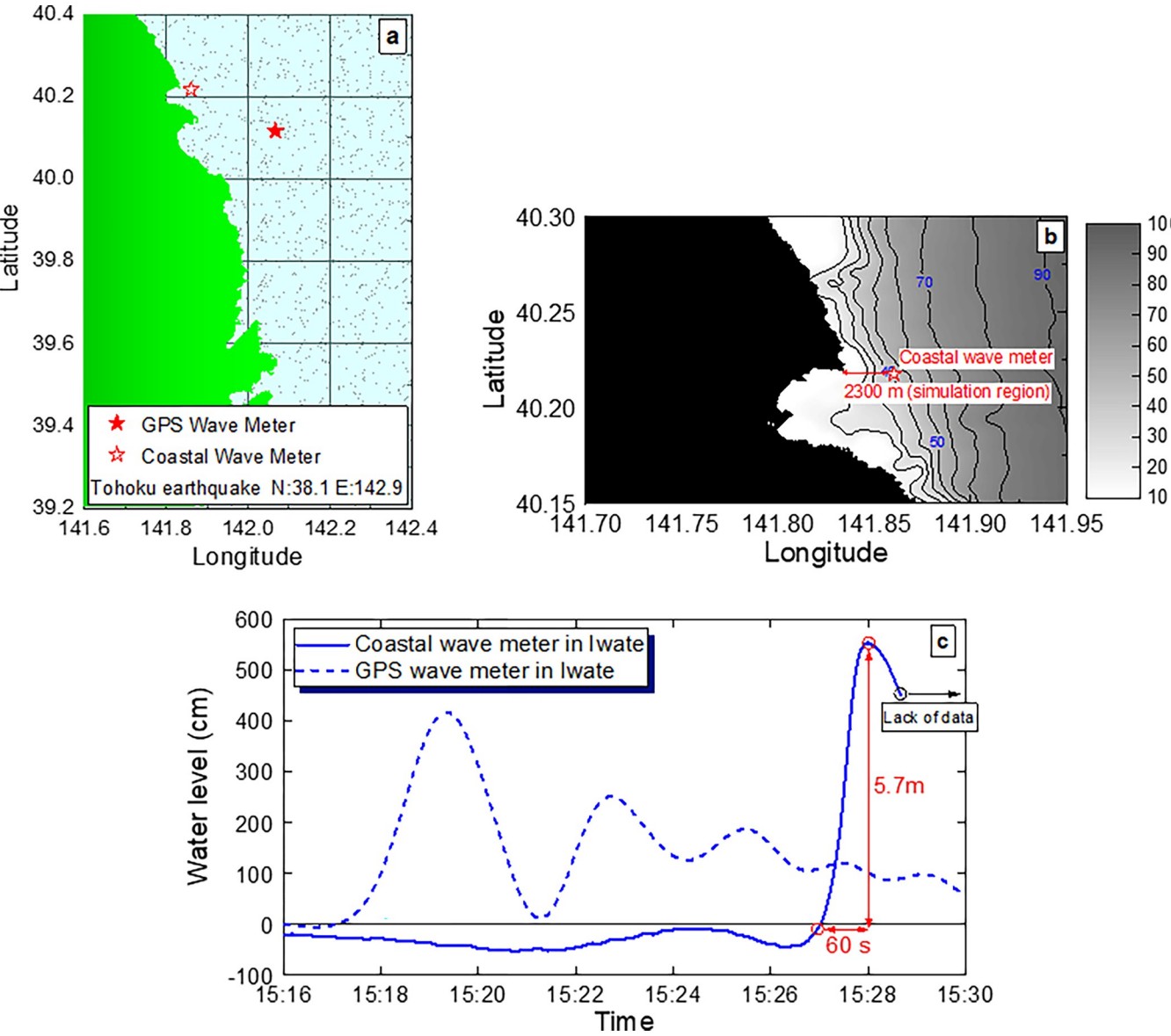

**Fig 2. Information on the Great East Japan Earthquake used to determine simulation conditions. a** Installation locations of the GPS wave meter and coastal wave meter off the coast of Iwate Prefecture. **b** Bathymetric data at the location where the coastal wave meter is installed. We obtained the 500 m mesh data from JODC (Japan Oceanographic Data Center) [17]. The water depth of 40 m at the position where the coastal wave meter was installed is taken as the water depth at the wave generation position. **c** Variations in water level observed off the coast of Iwate Prefecture on March 11, 2011. We obtained the data from NOWPHAS (Nationwide Ocean Wave information network for Ports and HArbourS) [18]. In this study, to reduce the computational load, the observed data from the coastal wave meter (blue solid line), which have a short period of water level change, were used as the tsunami waveform.

methodologies, there might be a need to incorporate statistical analysis, given that tsunami conditions can vary probabilistically.

## Acquisition and estimation of the waveforms and topographic data required for numerical simulations

In this study, the tsunami waveform observed using the coastal wave meter, as shown by the solid blue line in **Fig 2(C)**, was reproduced as linear waves (small-amplitude waves and Airy

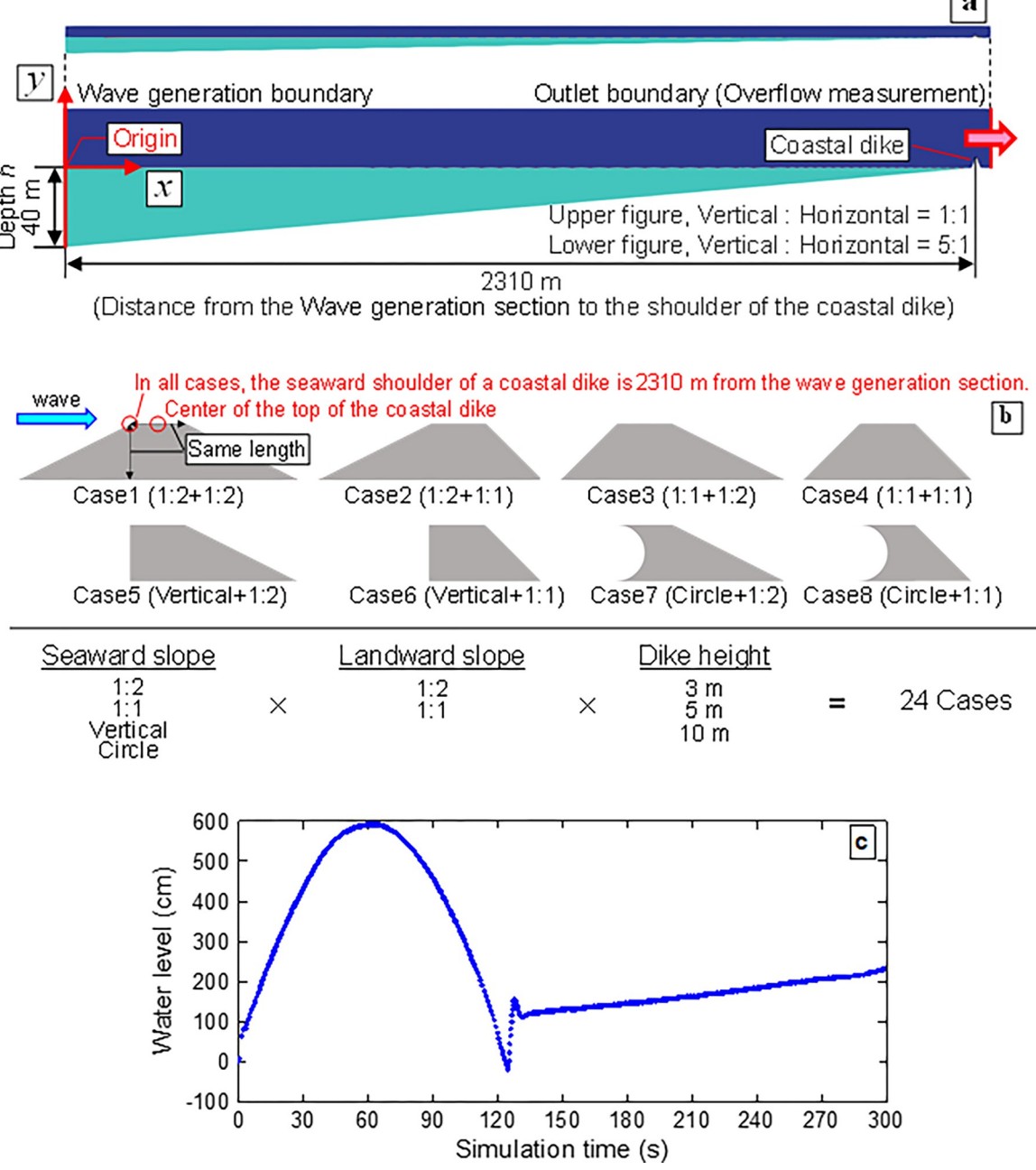

**Fig 3. Numerical simulation conditions. a** Simulation domain. For the simulation domain, the original scale and the 5-fold magnified scale in the y direction with enhanced visibility are illustrated. The left end of the domain is the wave-generation boundary (initial depth = 40 m), and the right end of the domain is the outlet boundary (overflow measurement). In all cases, the seaward shoulder of the coastal dike was set to x = 2310 m. **b** Dike shape. The reduction effects of 24 dike shapes on the total amount of overflow were examined. The circle on the seaward slope of the coastal dikes (Cases 7 and 8) was assumed to be the wave-return structure. **c** Tsunami waveform at x = 5 m near the wave-generation boundary in the numerical simulation.

waves). The linear wave is given by

$$\eta = \frac{H}{2}\cos(kx - \sigma t) \tag{1}$$

$$\phi = \frac{H}{2}\frac{\sigma}{k}\frac{\cosh k(y+h)}{\sinh kh}\sin(kx - \sigma t) \tag{2}$$

$$U_x = \frac{\partial \phi}{\partial x} = \frac{H}{2}\sigma\frac{\cosh k(y+h)}{\sinh kh}\cos(kx - \sigma t) \tag{3}$$

$$U_y = \frac{\partial \phi}{\partial y} = \frac{H}{2}\sigma\frac{\sinh k(y+h)}{\sinh kh}\sin(kx - \sigma t) \tag{4}$$

Eq (1) assumes that the water level change is a cosine wave. Eq (2) describes a potential flow when the amplitude is small compared to the wavelength without considering viscosity and vorticity, where $H$ is the wave height, $h$ is the water depth, $L$ is the wavelength, $T$ is the period, $k$ ($= 2\pi/L$) is the wavenumber, $\sigma$ ($= 2\pi/T$) is the angular velocity, $\phi$ is the flow velocity potential, $U_x$ is the velocity in the $x$ direction, and $U_y$ is the velocity in the $y$ direction. To apply Eqs (1), (3) and (4) as boundary conditions for water level change and flow velocity, it is necessary to determine the wave height $H$, water depth $h$, wavelength $L$ and period $T$. In this study, the data of the water level change observed by NOWPHAS (**Fig 2(C)**) is used as the wave height $H$ and period $T$, and the data from JODC (**Fig 2(B)**) is used as the water depth $h$. The wavelength $L$ is given by the product of the wave speed $c$ and the period $T$ of the long wave in the small-amplitude waves theory, described by the following equation.

$$cT = \sqrt{gh}T \simeq 4750(\text{m}) \tag{5}$$

From **Figs 2(B)** and **3(A)**, the water depth at the wave-generation boundary is 40 m, which gives a wave speed $c$ of approximately 19.8 m/s. Since the quarter period in **Fig 2(C)** is 60 s, the wavelength $L$ is 4750 m based on Eq (5).

## Details of numerical simulations

In this study, OpenFOAM [19,20] is used to perform the two-dimensional unsteady numerical simulations of the isothermal, incompressible and immiscible two-phase fluid flow (water and air). The finite volume method is used to solve the governing equations (continuity and momentum equations) given by

$$\frac{\partial \rho \boldsymbol{u}}{\partial t} + \nabla \cdot (\rho \boldsymbol{uu}) = -\nabla p + \nabla \cdot \left[\mu\{\nabla \boldsymbol{u} + (\nabla \boldsymbol{u})^T\}\right] - \nabla\left(\frac{2}{3}\mu\nabla \cdot \boldsymbol{u}\right) + \boldsymbol{F} \tag{6}$$

where $\rho$ is the density, $t$ is the time, $\boldsymbol{u}$ is the velocity vector, $p$ is the pressure, $\mu$ is the viscosity coefficient, and $\boldsymbol{F}$ is the external force vector. Assuming a temperature of 20°C and a pressure of 100 kPa, the density and kinematic viscosity coefficient of water are 998 kg/m$^3$ and $1.00 \times 10^{-6}$ m$^2$/s$^2$, and those of air are 1.19 kg/m$^3$ and $1.53 \times 10^{-5}$ m$^2$/s$^2$. The PISO (Pressure Implicit with Splitting Operators) [21], $k$–$\varepsilon$ model [22] and VOF (Volume Of Fluid) [23] methods were used for pressure–velocity coupling, the turbulence model and tracking the gas–liquid interface, respectively. The advection equation for the VOF method in OpenFOAM is given by

$$\frac{\partial \alpha}{\partial t} + \nabla \cdot (\alpha \boldsymbol{u}) + \nabla \cdot \{\alpha(1 - \alpha)\boldsymbol{u}_{\mathbf{r}}\} = 0 \tag{7}$$

where $\alpha$ is the volume fraction of the liquid ($0 \leq \alpha \leq 1$), and $\boldsymbol{u}_{\mathrm{r}}$ is the relative velocity obtained by subtracting the air velocity from the water velocity. The upwind scheme was used as the

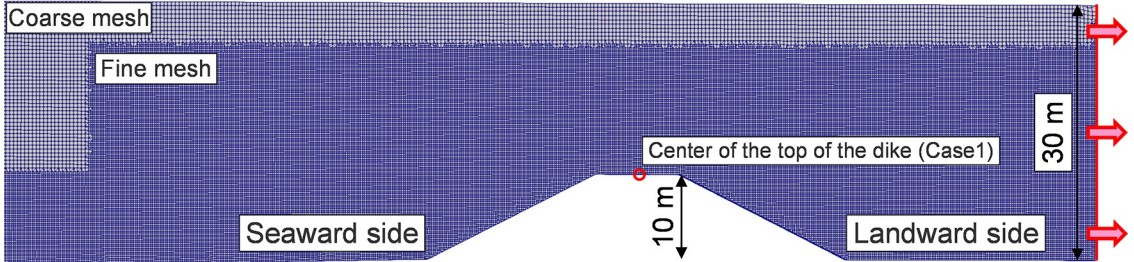

**Fig 4. Mesh around the coastal dike in Case 1 at the dike height of 10 m.** Smaller mesh sizes were applied to the space around the dike, which is critical for the tsunami overflow.

advection term for the velocity vector $u$, turbulence energy $k$ and energy dissipation rate $e$, and the vanLeer scheme was used for the advection term for the volume fraction $\alpha$. The total number of cells in Case 1 with the dike height of 3 m is 1132430. The mesh near the coastal dike in Case 1 at the dike height of 10 m is shown in **Fig 4**. A smaller mesh size is applied to the space around the coastal dike, which is important for tsunami overflow. Ryzen Treadripper 3990X was used as the CPU, and the number of parallelisms was set to 118. A variable time width was set for the calculation, where the Courant number did not exceed 0.7. The conditions for the numerical simulations are listed in **Table 1**.

## Validation of numerical simulations

To ensure the validity of numerical simulations, we compared our simulations with the experiment findings of dike overflow conducted by Mitsui et al. [16] as a preliminary validation. Note that the simulation results of the dike shape and overflow, which are the objectives of this study, are presented in the following Results section. In the experiment, the model with a dike height of 4 cm was subjected to a steady flow with an overflow depth of 9 cm, and the water surface height and velocity were recorded. As described in the above section, interFoam was used as the simulation solver, and the $k$–$\varepsilon$ model was used as the turbulence model. **Fig 5** shows the simulation conditions and results. The simulation area accurately reproduces the dike shape, although it does not fully reproduce the experimental dimensions as some of the experimental dimensions are unclear. **Fig 5** shows that the numerical simulation reproduces the experimental results well with respect to the water surface level overflowing the dike. The

**Table 1. Conditions of the numerical simulation in Case 1 with the dike height of 3 m.**

| | |
|---|---|
| Total number of cells | 1132430 |
| Solver | interFoam |
| Turbulence model | $k$–$\varepsilon$ model |
| Wave type | Airy waves |
| Water depth at the inlet boundary | 40 m |
| Wavelength | 4750 m |
| Courant number | 0.7 |
| Simulation time | 300 s |
| Boundary condition type for velocity | Inlet: waveVelocity<br>Outlet: inletOutlet |
| Boundary condition type for pressure | Inlet: zeroGradient<br>Outlet: zeroGradient |
| Boundary condition type for alpha | Inlet: waveAlpha<br>Outlet: zeroGradient |

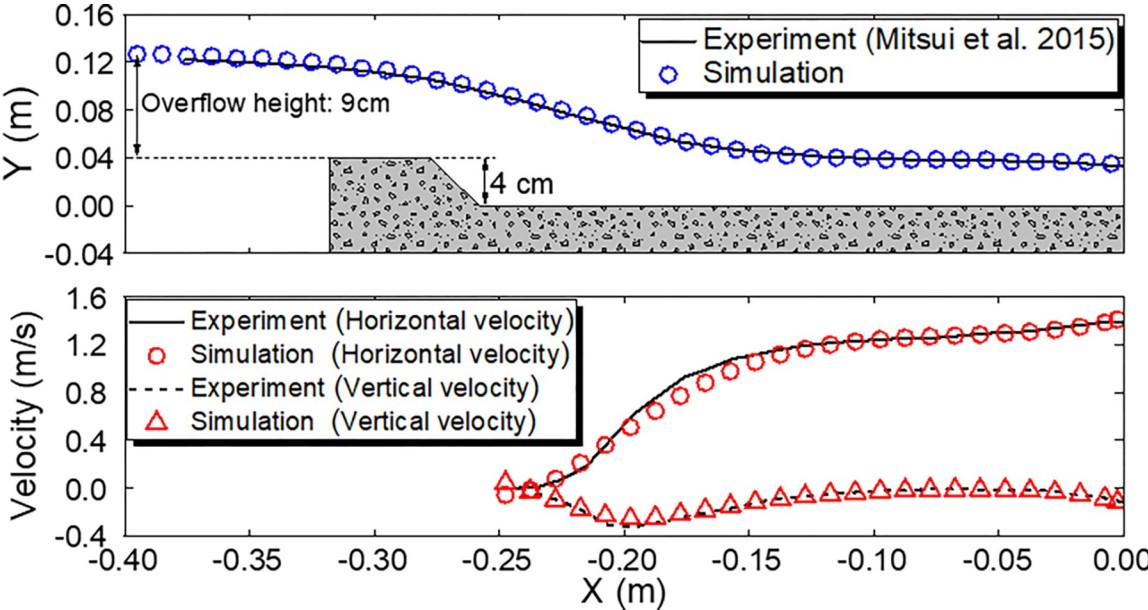

**Fig 5. Numerical simulation conditions and results. a** Water level distribution. By generating a constant volume of water from the bottom of the inflow boundary, steady flow conditions of an overflow depth of 9 cm are reproduced as reported by Mitsui et al. [16]. **b** horizontal and vertical velocity distributions on the landward side of the coastal dike. The experimental and numerical results are in good agreement regarding the water level and velocity distributions, confirming the validity of the simulation method used in this study.

numerical and experimental results of the velocity distributions in the *x* and *y* directions on the landward side of the dike also agree well, suggesting that the reproduction of the overflow phenomenon using the simulation method in this study is reasonable.

## Results

### Total amount of overflow for all 24 cases at the end of the simulation (after 300 s)

**Fig 6(A)** shows the variation in the instantaneous overflow rate over time at the outlet boundary for Case 2, which has the highest total amount of the overflow at the dike height of 5 m, and Case 7, which has the lowest total amount. **Fig 6(B)** shows the variation in the accumulated amount of the overflow over time. **Fig 6(C)** shows the total amount ratio of the overflow for each dike shape at each dike height (3 m, 5 m and 10 m). The total amount ratio of the overflow decreases by a maximum of 6.5% at the dike height of 3 m, 11.0% at the dike height of 5 m, and 30.0% at the dike height of 10 m. The decrease in the total amount of the overflow at each dike height indicates that the total amount varies depending on the dike shape and that disaster mitigation by reducing the total amount can be expected by selecting an appropriate dike shape. As for the influence of the seaward slope on the total amount ratio of the overflow, the total amount decreases in the order of 1:2 (Cases 1 and 2), 1:1 (Cases 3 and 4) and Vertical (Cases 5 and 6) at each dike height, indicating that steepening the seaward slope is effective in reducing the overflow. The wave-return structures in Cases 7 and 8 are also highly effective in the overflow reduction. Although wave-return structures have been confirmed to be effective for waves with short wavelengths [24–26], their effectiveness for long-period waves such as tsunami has not been previously reported. Therefore, the findings of this study are extremely valuable.

Comparing the cases with different landward slopes at each dike height, the maximum difference in the total amount ratio of the overflow is 1.7% (Cases 7 and 8 at the dike height of 5

m). The total amount ratio of the overflow is smaller with the 1:1 landward slope at the dike height of 3 m, whereas it is smaller with the 1:2 landward slope at dike heights of 5 m and 10 m. This is due to the tsunami overflow conditions, which is determined by the relationship between the dike height and tsunami scale. The maximum reduction ratio of the total amount of the overflow increases as the dike height increases because the flow pattern at the initial stage of the overflow changes significantly depending on the dike height. For instance, when the dike height is 10 m, the return flow of the tsunami back to the seaward side is formed in Vertical (Cases 5 and 6) and Circle (Cases 7 and 8). The detailed flow pattern is described in the next section.

## Detailed flow conditions and reasons for the decrease in the total amount of the overflow

To examine the effects of different seaward slopes of coastal dikes (1:2, 1:1, Vertical and Circle) on the total amount of the overflow, Fig 7 shows the flow pattern during the tsunami overflow. Fig 7(A) shows the water level change at the center of the top of the coastal dike (see Case 1 in Fig 3(B)) for Cases 1, 3, 5 and 7 at each dike height. At the initial stage (from 155 s to 175 s), the water level change differs depending on the dike shape. The return flow of the arriving tsunami to the seaward side is formed in Case 7 at the dike height of 10 m (Fig 7(A)), and the start time of the overflow is much later than those observed in the other cases. For the dike heights of 5 m and 10 m the start time of the overflow is delayed in Cases 5 and 7 compared to Case 1, whereas, for the dike height of 3 m, the start time of the overflow is similar regardless of the dike shape due to the insufficient dike height. The delay in the start time of the overflow due to the return flow contributes to the reduction in the total amount of the overflow observed in Cases 5–8 at dike heights of 5 m and 10 m, as shown in Fig 6(C). After the initial stage of the overflow, the water level at the center of the top of the coastal dike is independent of the dike shape at all dike heights.

Fig 7(B) shows the horizontal velocity distribution at the center of the top of the coastal dike for Cases 1 and 7 at each dike height 175 s after the start of the overflow. Comparing the horizontal velocity distributions for Cases 1 and 7 at each dike height, the horizontal velocity for Case 7 is smaller than that for Case 1 at all vertical heights. In particular, this velocity reduction effect is more pronounced at vertical heights between 0 m and 2 m (the blue region in Fig 7(B)). Fig 7(C) and 7(D) show the velocity distributions for Cases 1 and 7 at the dike height of 5 m after 175 s. As shown in Fig 7(D), in Case 7, the flow velocity decreases near the seaward top of the coastal dike compared to Case 1 (Fig 7(C)) because the flow tends to separate at the seaward top of the coastal dike when the seaward slope is shaped like Vertical or Circle.

The effect of different landward slopes (1:2 and 1:1) of the coastal dike on the total amount of the overflow is smaller than the effect of different seaward slopes. The largest difference in the total amount ratio of the overflow between the 1:2 and 1:1 landward slopes is 1.7% (Cases 7 and 8 at the dike height of 5 m) for the same seaward slope. The total amount of the overflow is smaller with the 1:1 landward slope than that with the 1:2 landward slope at the dike height of 3 m, whereas it is smaller with the 1:2 landward slope than that with the 1:1 landward slope at dike heights of 5 m and 10 m (Table 2). If the landward slope is considered as an enlarged conduit like a diffuser, the head loss [27] is smaller with the landward slope 1:2 (slope angle: 26.6˚) than with the 1:1 landward slope (slope angle: 45˚). Therefore, the total amount of the overflow is smaller with the 1:1 landward slope at the dike height of 3 m. At dike heights of 5 m and 10 m, the total amount of the overflow is smaller with the 1:2 landward slope, although the head of water loss at the widened area is smaller with the 1:2 landward slope because the

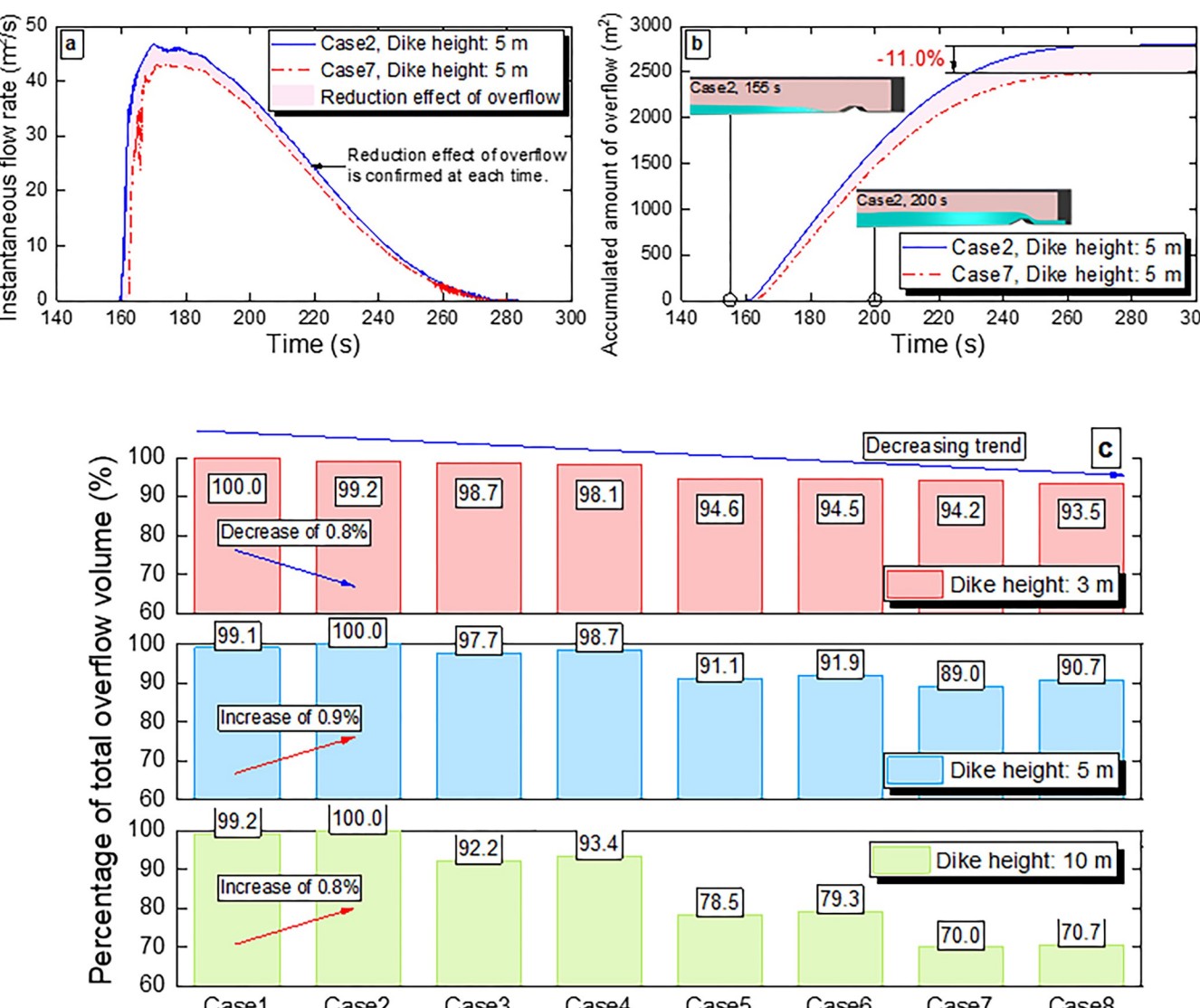

**Fig 6. Numerical simulation results for the amount of tsunami overflow. a** Variation in the instantaneous overflow rate over time for Cases 2 and 7 at the outlet boundary with the dike height of 5 m. At the dike height of 5 m, Case 2 has the highest total amount of the overflow, while Case 7 has the lowest total amount of the overflow. **b** Variation in the accumulated amount of the overflow over time for Cases 2 and 7 at the outlet boundary with the dike height of 5 m. **c** Total amount ratio of the overflow for all 24 cases at the end of the simulation (after 300 s). The case with the highest total amount ratio at each dike height (3 m, 5 m and 10 m) is set to 100%.

overflow conditions [28,29] are different from those at the dike height of 3 m. **Fig 8(A)** shows the overflow conditions at dike heights of 3 m and 10 m 175 s after the start of the overflow. At the dike height of 3 m, the stream curvature at the landward shoulder of the coastal dike is small because the dike height is small relative to the tsunami scale. On the contrary, at the dike height of 10 m, the stream curvature is large, and negative pressure is generated at the landward shoulder of the coastal dike (**Fig 8(B)** and **8(C)**). This negative pressure is greater with the 1:1 landward slope, where the curvature of the flowline is smaller, creating a forward pressure gradient in the direction of the flow at the top of the coastal dike. Therefore, the total amount of the overflow increases with the 1:1 landward slope. As summarized in **Table 2**, when the stream curvature is small, the head loss at the landward shoulder is the dominant

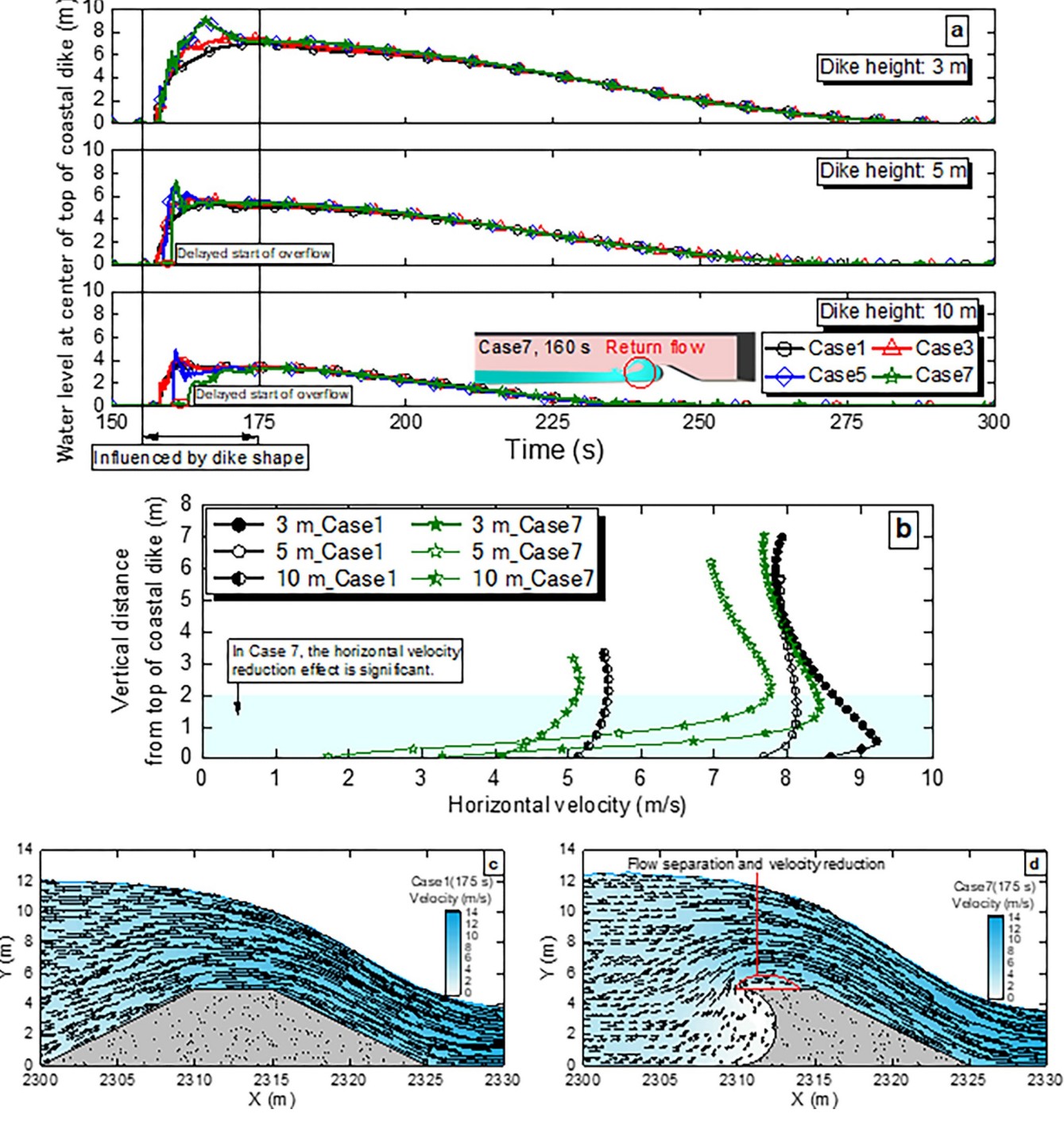

**Fig 7. Influence of the seaward slope of the coastal dike on the flow pattern. a** Variation in water level over time for Cases 1, 3, 5 and 7 at the center of the top of the coastal dike. The location of the center of the top of the coastal dike is shown in Case 1 of **Fig 3(B)**. The delay at the start of the overflow is observed in Cases 5 and 7 at dike heights of 5 m and 10 m. **b** Horizontal velocity distribution for Cases 1 and 7 at the center of the top of the coastal dike at each dike height (3 m, 5 m and 10 m) 175 s after the start of the overflow. the velocity reduction effect is observed for Case 7 at vertical heights from 0 m to 2 m. **c** Velocity distribution for Case 1 at the dike height of 5 m 175 s after the start of the overflow. **d** Velocity distribution for Case 7 at the dike height of 5 m 175 s after the start of the overflow. Compared to Case 1 (**Fig 7(C)**), the velocity for Case 7 is reduced due to flow separation at the seaward top of the coastal dike.

**Table 2. Influence of the dike height and the landward slope on the total amount of the overflow.**

| Dike height | Landward slope where the total amount of the overflow is smaller | Stream curvature | Dominant factor |
|---|---|---|---|
| 3 m | 1:1 (45.0˚) | Small | Head loss at the dike shoulder |
| 5 m and 10m | 1:2 (26.6˚) | Large | Negative pressure at the dike shoulder |

factor in the total amount of the overflow, reducing the overflow with the 1:1 landward slope. When the stream curvature is large, the dominant factor is the negative pressure generated at the dike shoulder due to overflow, reducing the overflow with the 1:2 landward slope.

## Discussion

### Effectiveness of dike shapes against long-period tsunamis

A significant question regarding this study is the extent to which the results and findings obtained can be applied to other tsunamis. We discuss the effectiveness of the dike shape for tsunami with periods longer than that reproduced in this study. To reiterate, the effect of different landward slopes on the total amount of the overflow is smaller than the effect of different seaward slopes. Therefore, the focus of the discussion on the effectiveness of the dike shape is primarily on the seaward slope. The reduction mechanism of the amount of the overflow due to the seaward slope identified in this study can be classified into two categories. The first mechanism is that the tsunami is reflected by the seaward slope of the coastal dike, delaying the start of the overflow. The second mechanism is that the flow velocity at the top of the coastal dike is reduced with dike shapes such as Vertical (Cases 5 and 6) and Circle (Cases 7 and 8).

The delay at the start of the overflow can be clearly seen in Case 7 at the dike height of 10 m. As a result, the total amount ratio of the overflow is approximately 30% lower than that for

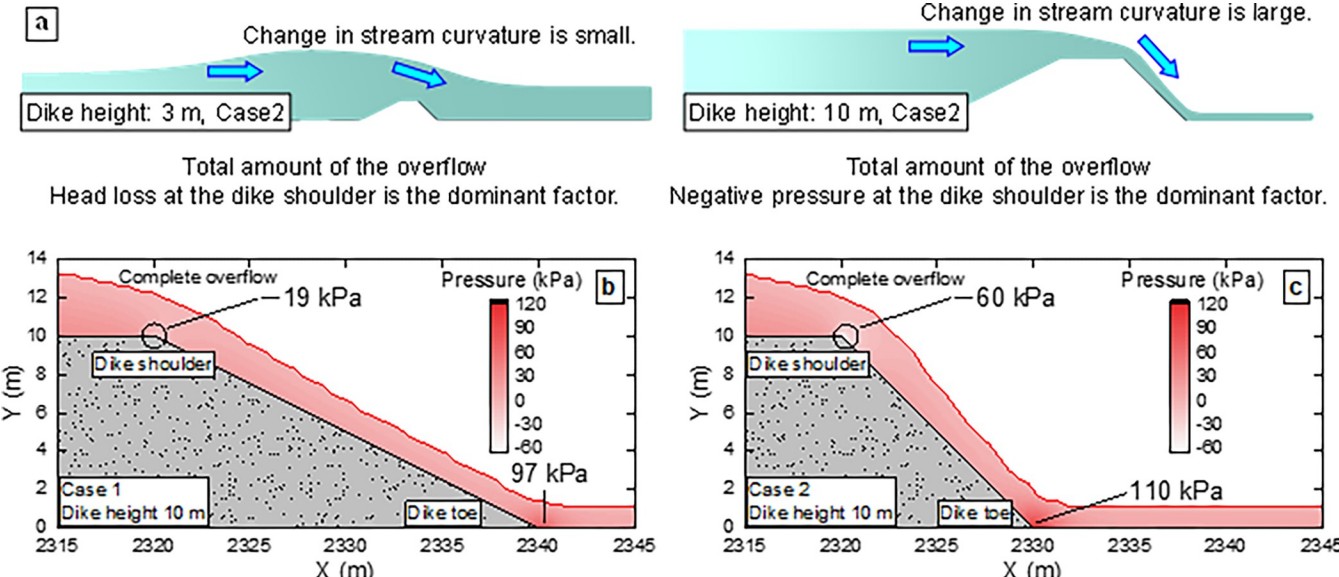

**Fig 8. Water level and pressure distributions at dike heights of 3 m and 10 m after 175 s. a** Overflow conditions in Case 2 at the 3 m (small stream curvature) and 10 m (large stream curvature) dike heights 175 s after the start of the overflow. **b** Pressure distribution of Case 1 at the dike height of 10 m after 175 s. **c** Pressure distribution of Case 2 at the dike height of 10 m after 175 s. In Case 2, compared to Case 1, negative pressure (forward pressure gradient) develops on the slope shoulder, and the pressure increases at the slope toe.

Case 2. Since the reduction effect due to the delay at the start of the overflow is exerted in the initial stage of the overflow, this effect is expected to become smaller as the wave period of tsunami becomes longer. Therefore, a reduction of approximately 30% in the total amount ratio of the overflow cannot be expected for tsunami with a longer period than that used in this study. However, from the viewpoint of securing the evacuation time [30,31], not from the viewpoint of reducing the total amount ratio of the overflow, the delay at the start of the overflow is an important point for reducing tsunami damage regardless of the wave period of tsunami. On the other hand, the reduction in the total amount of the overflow due to the reduced velocity at the top of the coastal dike is considered to be effective, even for long-period tsunami. As shown in **Fig 6(A)**, the instantaneous flow rate for Case 7 at the dike height of 5 m confirmed at the outlet boundary of 5 m is always lower than that for Case 2, indicating that the reduction effect is not limited to the initial stage of the overflow. As can be seen from the flow distribution at the top of the coastal dikes shown in **Fig 7(B)**, the velocities at each dike height are reduced by applying the circular dike shape (Case 7). Since no delay at the start of the overflow due to the reflection of tsunami is observed at the dike height of 3 m, the reduction in the total amount of the overflow can be attributed to the reduction in flow velocity at the top of the coastal dike, and the total amount ratio of the overflow at a dike height of 3 m is reduced by a maximum of 6.5%, as shown in **Fig 6(C)**. Therefore, the same level of reduction can be expected for longer-period tsunami.

## Proposal for dike shape to reduce tsunami overflow

Based on the simulation results, a proposal for the dike shape to reduce tsunami overflow is presented. For the seaward slope of a coastal dike, a shape such as Vertical or Circle is desirable because it is more effective in reducing the total amount of the overflow at each dike height. Specifically, due to the two reduction mechanisms explained in the previous section, Vertical reduces the total amount ratio of the overflow by 5.4%–21.5% and Circle by 5.8%–30.0% in this study (**Fig 6(C)**). The optimum shape of the landward slope of a coastal dike depends on overflow conditions. When the dike height is sufficient for tsunami overflow (large stream curvature), the total amount of the overflow is smaller with the 1:2 landward slope; when the dike height is insufficient (small stream curvature), the total overflow is smaller with the 1:1 landward slope. When the largest class of tsunami overflow occurs, the dike height is considered insufficient for the tsunami scale. In addition to the effect of reducing the total amount of the overflow against a giant tsunami, the 1:1 landward slope has the advantage of using a smaller landward area compared to the 1:2 landward slope. However, while the influence of the landward slope on the overflow is complex, it should be noted that the largest difference in the total amount ratio of the overflow between the 1:2 and 1:1 landward slopes is 1.7%, indicating that its impact is minor. In the following section, we discuss other considerations for coastal dikes not addressed in this study.

## Important considerations for coastal dikes not covered in this study

This study primarily focuses on examining dike shapes that are effective in reducing the amount of tsunami overflow. While the dike shape is a critical factor for the overflow reduction, other aspects like the durability and stability of coastal dikes are equally important. Therefore, these aspects need to be verified separately. For the seaward slope, a dike shape such as Vertical or Circle effectively reduces the overflow. However, a substantial impact force is anticipated when a tsunami reaches the dike. For the landward slope, as shown in **Fig 8(B)** and **8(C)**, the 1:1 slope develops negative pressure at the landward shoulder of the coastal dike and increases pressure at the landward toe compared to the 1:2 slope. The negative pressure at the

dike shoulder causes the detachment of covering blocks. In addition, the scour on the landward ground of the coastal dike destabilizes the dike, and based on a field survey, Kato et al. [32] reported that the scour on the landward ground of the coastal dike caused 49.2% of dike failures. Since the largest difference in the total amount ratio of the overflow between the 1:2 and 1:1 landward slopes is 1.7% (Cases 7 and 8 at the dike height of 5 m), a gentle landward slope such as 1:2 is desirable when the dike stability is prioritized over the reduction of the total amount of the overflow by the landward slope. Iiboshi et al. [33], Mitobe et al. [34,35], Takegawa et al. [36–39] and Rahman et al. [40–42] proposed various scour countermeasures. Therefore, when installing a steep landward slope due to site constraints, it is desirable to suppress the scour due to tsunami overflow by referring to the above methods.

## Conclusions

### Key findings

➢ The seaward slope (1:2 (26.6˚), 1:1 (45˚), Vertical and Circle) of a coastal dike remarkably affects the total amount of the overflow.

➢ Using a Vertical or a Circle on the seaward slope, a 5%–30% reduction of the total amount of the overflow is achieved compared to the 1:2 seaward slope. The fact that the total amount of the overflow is reduced by up to 30% without changing the dike height is truly groundbreaking.

➢ The reasons for the overflow reduction related to the seaward slope are identified as (1) delay at the start of the overflow due to wave reflection and (2) reduced velocity at the top of the coastal dike.

➢ Comparing the cases in which the landward slope (1:2 and 1:1) varies, the maximum difference in the total amount of the overflow is 1.7%. The influence of the landward slope on the total amount of the overflow differs depending on the overflow condition, which is determined by the relationship between the dike height and tsunami scale.

### Recommendations

➢ The stability of coastal dikes needs to be verified separately because of the large wave forces acting on the vertical walls or wave-return structures.

➢ When the dike height is sufficient for tsunami overflow (large stream curvature), the total amount of the overflow is smaller with the 1:2 landward slope than that with the 1:1 landward slope. When the dike height is insufficient (small stream curvature), the total amount of the overflow is smaller with the 1:1 landward slope than with the 1:2 landward slope.

➢ While the influence of the landward slope on the overflow is complex, it should be noted that the largest difference in the total amount ratio of the overflow between the 1:2 and 1:1 landward slopes is 1.7%, indicating that its impact is minor. Therefore, a gentle landward slope such as 1:2 is desirable when the dike stability is prioritized over the reduction of the total amount of the overflow by the landward slope.

➢ The development of negative pressure at the landward slope shoulder and the increase in pressure at the landward slope toe are observed for the 1:1 landward slope compared to the 1:2 landward slope. Therefore, countermeasures are necessary for the detachment of covering blocks due to the negative pressure and the scour on the landward ground of coastal dikes.

## Supporting information

**S1 Dataset.**
(XLSX)

## Author Contributions

**Conceptualization:** Naoki Takegawa.

**Data curation:** Naoki Takegawa.

**Funding acquisition:** Yutaka Sawada.

**Methodology:** Naoki Takegawa.

**Supervision:** Yutaka Sawada, Noriyuki Furuichi.

**Validation:** Naoki Takegawa, Yutaka Sawada, Noriyuki Furuichi.

**Visualization:** Naoki Takegawa.

**Writing – original draft:** Naoki Takegawa.

**Writing – review & editing:** Naoki Takegawa, Yutaka Sawada, Noriyuki Furuichi.

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
