## [Decision Letter · Decision Letter 0]

15 Aug 2023

PONE-D-23-18971Strategic Coastal Dike Shape For Enhanced Tsunami Overflow ReductionPLOS ONE

Dear Dr. TAKEGAWA,

Thank you for submitting your manuscript to PLOS ONE. After careful consideration, we feel that it has merit but does not fully meet PLOS ONE’s publication criteria as it currently stands. Therefore, we invite you to submit a revised version of the manuscript that addresses the points raised during the review process. Please submit your revised manuscript by Sep 29 2023 11:59PM. If you will need more time than this to complete your revisions, please reply to this message or contact the journal office at plosone@plos.org. Please include the following items when submitting your revised manuscript:A rebuttal letter that responds to each point raised by the academic editor and reviewer(s). You should upload this letter as a separate file labeled 'Response to Reviewers'.A marked-up copy of your manuscript that highlights changes made to the original version. You should upload this as a separate file labeled 'Revised Manuscript with Track Changes'.An unmarked version of your revised paper without tracked changes. You should upload this as a separate file labeled 'Manuscript'.If applicable, we recommend that you deposit your laboratory protocols in protocols.io to enhance the reproducibility of your results. Protocols.io assigns your protocol its own identifier (DOI) so that it can be cited independently in the future. For instructions see: https://journals.plos.org/plosone/s/submission-guidelines#loc-laboratory-protocols. Additionally, PLOS ONE offers an option for publishing peer-reviewed Lab Protocol articles, which describe protocols hosted on protocols.io. Read more information on sharing protocols at https://plos.org/protocols?utm_medium=editorial-email&utm_source=authorletters&utm_campaign=protocols.

We look forward to receiving your revised manuscript.

Kind regards,

Ahmed Mancy Mosa, Ph.D.

Academic Editor

PLOS ONE

Journal Requirements:

"This work was supported by JSPS KAKENHI Grant Number 21H02306."

"This work was supported by JSPS KAKENHI Grant Number 21H02306 (S.Y. and T.N.).

https://www.jsps.go.jp/english/e-grants/ The funder had no role in study design, data collection and analysis, decision to publish, or preparation of the manuscript."

**Additional Editor Comments:**

Please consider all comments.

Reviewers' comments:

Reviewer's Responses to Questions

**Comments to the Author**

1. Is the manuscript technically sound, and do the data support the conclusions?

Reviewer #1: Yes

Reviewer #2: Yes

Reviewer #3: Yes

2. Has the statistical analysis been performed appropriately and rigorously? 

Reviewer #1: Yes

Reviewer #2: N/A

Reviewer #3: No

3. Have the authors made all data underlying the findings in their manuscript fully available?

Reviewer #1: Yes

Reviewer #2: Yes

Reviewer #3: Yes

4. Is the manuscript presented in an intelligible fashion and written in standard English?

Reviewer #1: Yes

Reviewer #2: Yes

Reviewer #3: Yes

5. Review Comments to the Author

Reviewer #1: Title: Strategic Coastal Dike Shape for Enhanced Tsunami Overflow Reduction

Manuscript Number: PONE-D-23-18971

Comments for improvement of the paper

Abstract should incorporate the problems under investigation, purpose, methodology used, finding and recommendations. However, this paper failed to include some major components. Therefore, missed components should be included.

Key words should be included after the abstract in the main manuscript.

The discussion part of the paper is shallow.

Since the study will have paramount significance recommendation should be included after the conclusion.

Reviewer #2: Dear Authors

Thanks for your very interesting and informative scientific study. There are a few minor problems that need to be addressed:

Abstract:

1- It is better to start the first part of the summary of the article with the problem statement and then state the objectives of the current study in order to solve the aforementioned problems.

2- Please consider rewriting keywords according to National Library of Medicine terms accessible at : https://meshb.nlm.nih.gov

Materials and Methods:

1- It is reasonable to describe numerical simulations as a scientific technique before stating it's use in this study at the first paragraph.

Discussion:

1- Please state the limitations of the current study and possible limitations or errors in using the desired technique. (If there is no limit, it should be mentioned.)

Reviewer #3: 1. There were no analysis statistical method used in the study. It should be explained clear in the Method.

2. All another studies had been explained clearly in the study.

3. The research should put the weakness of using dike shape.

6. PLOS authors have the option to publish the peer review history of their article (what does this mean?). If published, this will include your full peer review and any attached files.

Reviewer #1: **Yes: **Fentahun Gebrie Mucha

Reviewer #2: No

Reviewer #3: No

---

## [Author Response · Author response to Decision Letter 0]

19 Sep 2023

September 2023

Dr. Ahmed Mancy Mosa, Ph.D.

PLOS ONE

Dear Editor:

I have uploaded the revised manuscript, response to reviewers (with highlighted corrections), and additional supporting file.

Please refer to the "response to reviewers" and "revised manuscript" for the responses to the reviewers' comments.

Regarding the non-review-related points mentioned in the email body, I have addressed them as follows in bullet points:

I have removed the information (This work was supported by JSPS KAKENHI Grant Number 21H02306.) about funding from the manuscript.

Please retain the Funding Statement as it currently stands.

"This work was supported by JSPS KAKENHI Grant Number 21H02306 (S.Y. and T.N.). https://www.jsps.go.jp/english/e-grants/ The funder had no role in study design, data collection and analysis, decision to publish, or preparation of the manuscript."

For Data Availability, I have compiled the dataset used in this study into a single Excel file and uploaded it as a Supporting Information file. 

Additionally, in line with this, I've updated the Data Availability Statement and incorporated the Supporting Information within the manuscript.

I have also confirmed that the references are complete and correct.

If you have any questions or require further clarification, please feel free to contact me.

National Institute of Advanced Industrial Science and Technology (AIST)

National Metrology Institute of Japan (NMIJ)

TAKEGAWA Naoki

Tel: 029-862-6359, Address: Central 3, 1-1-1, Umezono, Tsukuba, Ibaraki

---

## [Decision Letter · Decision Letter 1]

2 Oct 2023

Strategic Coastal Dike Shape For Enhanced Tsunami Overflow Reduction

PONE-D-23-18971R1

Dear Dr. TAKEGAWA,

We’re pleased to inform you that your manuscript has been judged scientifically suitable for publication and will be formally accepted for publication once it meets all outstanding technical requirements.

Kind regards,

Ahmed Mancy Mosa, Ph.D.

Academic Editor

PLOS ONE

Additional Editor Comments (optional):

Reviewers' comments:

Reviewer's Responses to Questions

**Comments to the Author**

1. If the authors have adequately addressed your comments raised in a previous round of review and you feel that this manuscript is now acceptable for publication, you may indicate that here to bypass the “Comments to the Author” section, enter your conflict of interest statement in the “Confidential to Editor” section, and submit your "Accept" recommendation.

Reviewer #2: All comments have been addressed

Reviewer #3: All comments have been addressed

2. Is the manuscript technically sound, and do the data support the conclusions?

Reviewer #2: Yes

Reviewer #3: Yes

3. Has the statistical analysis been performed appropriately and rigorously? 

Reviewer #2: Yes

Reviewer #3: Yes

4. Have the authors made all data underlying the findings in their manuscript fully available?

Reviewer #2: Yes

Reviewer #3: Yes

5. Is the manuscript presented in an intelligible fashion and written in standard English?

Reviewer #2: Yes

Reviewer #3: Yes

6. Review Comments to the Author

Reviewer #2: Thanks to authors for considering all comments and the all needed corrections. I hope to see other scientific works in future.

Reviewer #3: The manuscript is good enough so I recommend to be accepted.

The novelty of research idea is quite good.

7. PLOS authors have the option to publish the peer review history of their article (what does this mean?). If published, this will include your full peer review and any attached files.

Reviewer #2: **Yes: **Dr. Reza Habibisaravi, MD, PhD

Reviewer #3: No

---

## [Editor Report · Acceptance letter]

5 Oct 2023

PONE-D-23-18971R1 

Strategic coastal dike shape for enhanced tsunami overflow reduction 

Dear Dr. Takegawa:

I'm pleased to inform you that your manuscript has been deemed suitable for publication in PLOS ONE. Congratulations! Your manuscript is now with our production department. 

Kind regards, 

on behalf of

Dr. Ahmed Mancy Mosa 

Academic Editor

PLOS ONE